# Curriculum Development of EdTech Class Using 3D Modeling Software for University Students in the Republic of Korea

Wonjae Choi and Seonggyu Kim *

Academy of Cultural Studies, Dongguk University-Seoul, Seoul 04620, Republic of Korea; jacknroll49@gmail.com
* Correspondence: rkensin@gmail.com

**Abstract:** This study discusses the development of a software-centered EdTech (Education Technology) class model via the implementation of a course titled "3D Time Machine" at a 4-year university in the Republic of Korea over two semesters. The course focused on teaching the 3D modeling software Blender within the history department. The primary objective of offering this course was to equip students from the digital generation with the capability to manipulate digital technology effectively for their sustainable lives and individual development. By studying historical materials and accumulating domain knowledge, students could construct their narratives from their unique perspectives. This aimed to foster their proficiency in digital technology operation, preparing them for a sustainable education environment increasingly centered around virtual worlds. As the use of virtual worlds gains prominence in educational settings, there is a growing need to incorporate curricula that prepare students to thrive in a "leaving no one behind" society as well-prepared citizens. Assessing the digital competencies of contemporary university students and designing instructional models with particular attention to their needs is becoming increasingly important. This research draws insights from interviews, conducted in both face-to-face and written formats, with students who participated in the "3D Time Machine" course. The interviews revealed valuable insights that can be actively incorporated into the development of software-centered EdTech instructional models. They reported that they expanded their perceptions as they learned how to make their ideas tangible. The course helped students overcome their fear caused by the vagueness of digital technology. These opinions significantly contribute to the development of practical digital educational courses that can be easily and rapidly acquired and applied within virtual educational environments. In conclusion, this kind of course effectively employs 3D modeling technology, a software-centered EdTech, as a core element in helping students develop their narratives rapidly and diversely, thereby playing a crucial role in their ability to articulate their unique perspectives.

**Keywords:** EdTech; 3D modeling; digital narrative; software-centered class; Blender; 3D design; 3D education; university education; digital education; higher education; sustainable education; SDGs

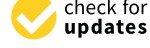



## 1. Introduction

### 1.1. Current Situation and Problem Statement of Sustainable Education

According to the U.S. Partnership for Education for Sustainable Development, education for sustainability is defined as a combination of content, learning methods, and outcomes that helps students develop a knowledge base about the environment, the economy, and society, in addition to helping them learn skills, perspectives, and values that guide and motivate them to seek sustainable livelihoods, participate in a democratic society, and live in a sustainable manner. A still-evolving field, sustainability education has the primary goal of harnessing the power of education to advance environmental literacy and civic engagement that prepares students for jobs that contribute to a more equitable and sustainable future [1].

In the current era marked by the rapid advancement of digital technology, everything is transitioning into a digital environment. Consequently, digital education has become an essential aspect for humanities major university students. Especially for these students, if they cannot integrate digital technology into their academic pursuits due to the boundaries and constraints of their majors, they will inevitably face difficulties adapting to the accelerating digital environment. In other words, for humanities major university students, digital education has become an indispensable element for a sustainable and evolving future.

In 2010, the German Rectors' Conference (Hochschulrektorenkonferenz, HRK) and the German Commission for UNESCO (Deutsche UNESCO-Kommission, DUK) issued a declaration urging universities to more vigorously pursue the ideal of sustainable development and to incorporate "education for sustainable development as a constructive element in all areas of university activity." [2,3] This means applying education for sustainable development in all areas of university activity, irrespective of the major, to enable students to lead sustainable lives both during their time at the university and after graduation. Therefore, it cannot be stressed enough that making digital technology education applicable to all fields of study and preparing students to navigate the digital landscape seamlessly is of paramount importance as a fundamental goal of university education.

"EdTech" (Education technology) that incorporates technologies such as AI, AR/VR/XR, and blockchain into traditional education is gaining significant attention. EdTech is now seen as a promising market with substantial potential due to the delayed digital transformation. The global education market, which serves as the fundamental environment for EdTech, is expected to grow from approximately 6.5 trillion dollars in 2020 to 8.1 trillion dollars in 2025 sooner or later, with projections indicating it will reach 10 trillion dollars by 2030 as shown in Figure 1. Within this, the EdTech market is expected to establish a market size of 342 billion dollars by 2025, up from 153 billion dollars in 2018 as shown in Figure 2. However, it is essential to note that EdTech's share in the overall education market remains relatively low. As of 2018, the proportion of the education market represented by EdTech was merely 2.5%. It is anticipated that by 2025, this proportion will only reach 4.3% [4]. This suggests that the digital transformation in the education sector is progressing relatively slowly compared to other sectors.

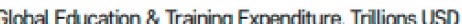

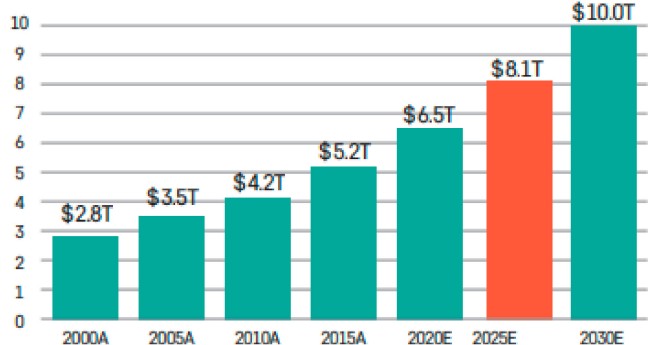

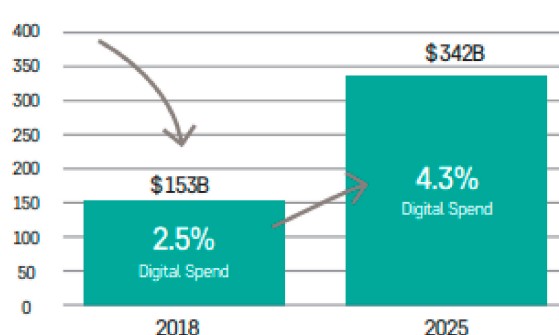

**Figure 1.** Global education and training market trend.

Figure 2 shows that the current issue in the EdTech market lies in the predominance of the hardware sector using AR/VR that produces standardized products. As the market for EdTech hardware grows, there is a lack of digital education curricula optimized for EdTech, which could foster learners who can actively create something using these tools. Consequently, learners were left with doubts about whether they needed to use EdTech in their personal study environments as opposed to the classroom setting. This hesitation has contributed to the sluggishness of the digitalization of education. It is in stark contrast to the rapid digital transformation pursued in the industry. Furthermore, the fact that those

currently leading the industry are not from the EdTech generation serves as another factor slowing the digital transformation of education.

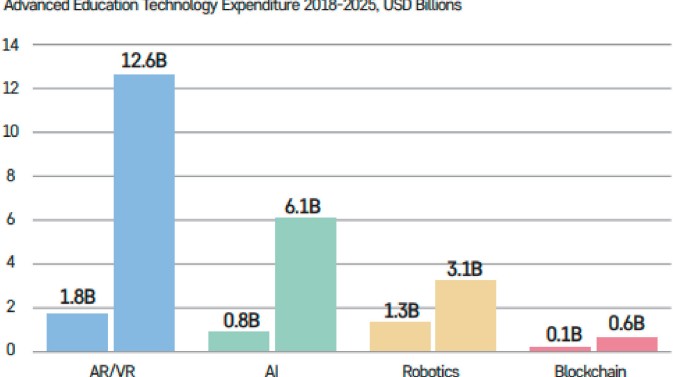

**Figure 2.** Advanced education technology expenditure trend.

Given this trend, there is concern that the future educational environment driven by EdTech may transform into a massive market dominated exclusively by a few technologists or electronic and big tech companies that have shaped the current EdTech market. It is more important to develop curricula specifically designed to utilize EdTech hardware to offer an education different from the past, led by educators of the humanistic-technologist type and equipped with a different perception to cater to new generations. To achieve this, organizations like "EdTech Impact" in the UK, "EdTech Evidence Exchange" in the USA, and "Education Alliance Finland" in Finland [5–8] have emerged as meta-EdTech organizations [9]. If the curriculum remains unchanged from past methods, while hardware evolves into EdTech, it will merely be a sophisticated "Edu analog" for transmitting knowledge in the traditional manner.

### 1.2. Digital Education of the Digital Generation for Sustainable Study and Life

Universities, as higher education institutions, hold various resources, including knowledge, human resources, physical facilities, and economic assets. They play a vital role in leading the localization of sustainable development objectives, addressing community issues, improving the welfare of local residents, and fostering an understanding of the goals of sustainable development in the community [10]. Universities are expected to play a crucial role in creating a more sustainable society via their roles in education and research. They contribute to nurturing integrated and systematic thinking abilities, which are essential for resolving issues that hinder sustainable development. Consequently, universities are fundamental places of learning, enabling students to make meaningful contributions to sustainable development via their academic courses.

Moreover, universities conduct research to discover solutions to urgent and significant problems that hinder sustainable development in our society. Therefore, universities serve as places where ideas are exchanged and new knowledge is created. Within universities, the active participation of members of the academic community fosters a mutual learning process between the university and society.

In the 55th United Nations General Assembly held in New York in September 2000, the "Millennium Development Goals (MDGs)" were adopted as an agenda. In June 2012, the United Nations Conference on Sustainable Development, known as the Rio + 20 Summit, took place in Rio de Janeiro, Brazil, where the "The Future We Want" declaration reaffirmed the commitment to sustainable development. This event led to an agreement on the "Sustainable Development Goals (SDGs)" in the international community. In the 70th United Nations General Assembly in September 2015, it was resolved to implement the Sustainable Development Goals from 2016 to 2030, succeeding the Millennium Development Goals that had expired in 2015 [11]. In this context, universities bear the responsibility to provide

education that enables the achievement of Sustainable Development Goals and promotes innovation.

In an era where the digital environment accelerates change, the responsibility to protect the environment and conserve resources is not limited to a few individuals but is a shared duty for all of humanity. Therefore, universities must empower their students to voluntarily work toward the SDGs, enabling them to live an eco-friendly academic and personal life within the digital landscape. Above all, current university students born after 2000 have grown up in a digital environment alongside digital devices. They must be seen as a generation capable of effectively utilizing digital technology as part of their daily lives and, thus, as a generation that can navigate the present and future effectively.

The era of distinguishing between the digitally literate generation and the digitally marginalized generation based on their proficiency in utilizing digital technologies has arrived. It is therefore reasonable to refer to this cohort as the digital generation. The survey results revealed that, based on the digital index, there are post-digital generation, digital generation, semi-digital generation, and analog generation [12]. And according to a recent "JIM" study in Germany, as many as 91 percent of teenagers play digital games, emphasizing the considerable influence digital games can have on school children [13].

Most of today's university students were born after the 2000s and have reached an age where they will soon enter society. Concerning EdTech and digital education, they are the primary targets. However, from the perspective of this research, their computer proficiency does not appear to be particularly outstanding.

According to the 2020 survey on the status of information technology in elementary and middle school education, digital generation students and the teachers who instruct them exhibit very basic levels of acceptance and utilization of information technology in education [14]. With the emphasis on cultivating EdTech utilization skills for prospective teachers amid "Education Innovation 4.0" and the digital transition, teacher training institutions responsible for teacher education have also expanded EdTech use and developed and operated EdTech-based educational programs. Nevertheless, teacher training institutions' EdTech utilization education lacks a comprehensive classification system for EdTech, and there is no systematic analysis of educational program operation [15].

In other words, compared to previous generations, these individuals have been exposed to digital devices more frequently and are familiar with handling them; however, it cannot be considered that they have smoothly received digital transition education. For example, they may have learned computer coding, but they do not actually apply it to their self-study or daily life. The only place where they use a computer for coding is outside the coding classroom. Digital education is not being utilized, and it only exists as an educational program for learning digital skills. Classes in other subjects are still conducted using traditional study methods.

No matter how much EdTech transforms the classroom into a digital environment, it has not been able to create a new educational paradigm exclusive to EdTech. Regarding this situation, Jewon Park mentioned, "The current classroom situation at schools remains at the second stage of understanding, and even this is overwhelming. AI can replace knowledge and understanding with adaptive learning, making school education unnecessary. Therefore, teachers need to change their roles so that students engage in third-stage creative learning, where they can apply and create." [16] This suggests that the paradigm of Korean elementary, middle, and high school education needs to change.

Ironically, the digital literacy level of the digital generation is low. In 2021, the Korean Educational Broadcasting System (EBS) conducted a literacy assessment of 2400 third-year middle school students, revealing that 27% of them could not comprehend textbooks, and 11% had the literacy level of elementary school students. In the 2018 Program for International Student Assessment (PISA) in reading, organized by the Organization for Economic Co-operation and Development (OECD), students who scored below level 2, which is considered subpar, accounted for 34.7%, while those at the basic literacy level, scoring level 1 or below, were 15.1%. Even the 2021 PISA report titled 21st Century

Readers: Developing Literacy in a Digital World showed that critical thinking about internet information was below the OECD average. Moreover, 15-year-old Korean students (middle school third-year and high school first-year) ranked among the lowest groups, along with countries like Mexico, Brazil, Colombia, and Hungary, when it came to identifying phishing emails. Furthermore, students who could not differentiate between facts and opinions were only 25.6% in Korea, significantly below the 47% average identification rate among OECD member countries [17].

However, there is still a strong demand for digital transformation education. In 2022, a survey conducted by the Korean Economic Daily in collaboration with the research firm Ipsos asked 300 parents from January 17 to 20 about the importance of AI and coding education compared to traditional subjects like Korean, English, and math. The results showed that 41.3% of respondents considered it "more important", and 11.3% regarded it as "very important." When combined with the "similar" response (40.3%), 93% of parents perceived digital-related AI and coding as essential subjects. Moreover, 92.6% of parents expressed their willingness to provide their children with paid or free AI and coding education [18]. Although the perception of the importance of digital literacy is growing, only 36.4% of parents were "satisfied" with the quality of information technology education in frontline schools, with 63.7% choosing "average" or "dissatisfied." Even university students recognize the importance of AI. In a survey of 150 university students, 57.3% of respondents stated they would seek extracurricular education to enhance their digital skills. Among these students, those in humanities and social sciences recorded a higher percentage at 63.9%, compared to students in natural and engineering sciences (56.3%). The conclusion drawn is that while digital skills are in demand outside the school, they are not adequately nurtured within the school environment. Soonmin Bae, the Director of the AI2XL Research Institute at KT Convergence Technology Institute, evaluated the situation in the education sector as being at a critical level [19].

To truly foster students' digital literacy, it is essential to contemplate whether expanding the digital education infrastructure and training more educators is indeed the right solution. Linguist Naomi S. Baron conducted three surveys involving over 10,000 university, high school, and middle school students. When asked, "What medium is most conducive for concentration and immersion?", 86% of university students and 85% of high school students chose printed books [20]. Similarly, when questioned about the most effective medium for learning and retention, 72% of university students and high school students favored printed books. These results alone underscore that digital application in education is not a panacea.

It becomes evident that the medium students find most effective for learning is not digital but rather print, suggesting that it is not enough to merely impose a digital framework on teaching analog subjects. What is needed is an educational approach that utilizes digital tools to teach in a way that was not possible before, revealing previously unexplored areas of knowledge and allowing students to learn beyond the conventional methods.

*1.3. Purpose*

A subsequent phase is the growing interest and presence of the metaverse, which, in a way, deepens the digital dimension in social and economic life and, potentially, in education itself in the (possible) development of the United Nation's Sustainable Development Goal 4—quality education. However, the metaverse as a learning environment is a topic that is (still) very little studied [20]. The metaverse is a new digitally implemented realm and space. As a result, it is essential to contemplate how we can effectively educate individuals on ways to thrive within the metaverse.

Therefore, this research aims to elucidate to what extent the generation commonly referred to as "digital natives"-the university students of today in 2023-are utilizing digital technology in their learning processes and what factors should be considered when designing effective curricula for them. Numerous studies, even in reputable media outlets, have been conducted on this subject. For instance, Donga Ilbo conducted an extensive analysis of

high school graduates from the year 2000 and summarized their characteristics into five categories, defining them as "smartphone-converged individuals," who are inseparable from their smartphones 24/7 [21]. As this result suggests, digital devices are integral to their daily lives. Therefore, it is generally assumed that the digital generation is well-acquainted with EdTech utilizing digital technology. However, based on the experiences of the author, who teaches at a university, these so-called "digital natives" do not particularly stand out when it comes to their digital proficiency.

The revised information education curriculum implemented in 2015 for middle and high school students was designed to enhance creative problem-solving skills, logical thinking, information culture literacy, and computational thinking. This generation was expected to produce creative, interdisciplinary talents with computational thinking skills according to the curriculum. However, in reality, it is challenging to consider them significantly different from the analog generation. They are essentially consumers of digital tools who merely use various digital devices. Therefore, this research aims to provide insight into what digital education means to this generation, what aspects they value, what they wish to learn, and how they view digital education and digital learning environments, contributing to the development of a digital literacy curriculum. Considering the current era in which education is increasingly conducted in virtual worlds, this study holds significance in providing an opportunity to assess what EdTech should encompass in the context of next-generation digital literacy education in virtual environments.

*1.4. Method*

After introducing the "3D Time Machine" course created by the author, this study examines the assignment approach adopted by the participating students and shares the results they achieved in the course. Via this study, it was possible to gauge the extent of students' digital growth and substantiate the educational validity of the "3D Time Machine" course. However, this study employs an interview and content analysis, qualitatively analyzing students' opinions on their experiences in the "3D Time Machine" course, gathered via students' written statements and a semi-structured face-to-face interview to find a thick background for a realistic digital education curriculum. All students submitted their opinions on the class freely after the session, and face-to-face interviews were conducted with students who wanted to express further opinions. While there were outstanding students in the 3D modeling assignment, proficiency in 3D modeling was not the criterion for participation in the interview.

A total of 20 students participated in this course over two semesters. They were all born after 2000 and majored in subjects such as history, history education, Korean literature, Buddhist studies, and law. Despite their self-reported limited knowledge of computers, particularly with no prior experience in 3D software, nine of them were interviewed. Face-to-face interviews took place in a free atmosphere for two hours after the course concluded, and statements made by students during the course were included in the content of the face-to-face interview. The interview consisted of reflections submitted via email after completing the course. In the face-to-face interviews, we posed the following questions:

1. What motivated you to enroll in the "3D Time Machine" course?
2. What aspects of the learning process did you find engaging? What were the reasons behind this?
3. What were the challenging aspects of the learning process? What were the reasons behind this?
4. What steps have you taken to learn digital literacy thus far?
5. Do you consider digital literacy when making career decisions?
6. Have there been any changes in your career aspirations after the course?
7. What positive aspects have you experienced after the course?
8. What aspects did you find lacking or disappointing after the course?

The opinions submitted by students in writing were extracted and analyzed using "qualitative content analysis," while face-to-face interviews were subjected to "thematic analysis" to identify themes and patterns in a more detailed manner.

Our study benefited from previous studies that employed methodologies similar to ours. For instance, Martin Krajcovic and Gabriela Gabajová conducted a study measuring students' proficiency levels after conducting two months of practical sessions handling engineering equipment in a virtual space [22]. Thanh Tuan To and Abdullah Al Mahmud analyzed the current status of educators' and university students' experiences with 3D printer usage and their associated challenges via surveys and interviews conducted at the university level [23]. Yu Fu and Hao Jiang performed a study involving kindergarten to fourth-grade students, who designed toy characters using a 3D cartoon maker, examining both the quality of the produced toys and student interviews [24]. There are studies analyzing the impact of educating workers in virtual reality in industrial settings or evaluating outcomes after educating students on artificial intelligence and digital technologies [25,26]. Most of these studies adopted methodologies involving practical classes over specific periods, followed by measuring changes in students or organizing interviews.

Our study is significant in its focus on university students who have not received any digital technology education in the context of South Korea's educational environment, particularly targeting those from elementary to high school. The study involved in-depth analysis of the 3D modeling designs created by students and conducted interviews, marking the significance of the study.

## 2. Developing an EdTech-Enhanced 3D Modeling Software Teaching Model

### 2.1. "3D Time Machine" Course

We contemplated the structure of virtual worlds via three key elements: "cognitive psychology," "design thinking," and "digital data narrative education." Especially the term "digital narrative" here refers to the skills of interpreting past materials existing solely as written records or preserved as images or architectural drawings and inputting them into 3D modeling software. It encompasses more than just acquiring knowledge from past materials; it includes the skill of reproducing them in virtual reality, transcending the mere acquisition of knowledge from historical data to the technology of connectivity in a virtual space.

The conductors introduced a major elective course titled "3D Time Machine" in the Department of History at "the A university" in the first semester of 2022. This course is a blend of "theoretical learning," "craftsmanship," and "practice" in the metaverse, designed to stimulate student engagement via primary and secondary source materials, allowing for the generation of innovative content via the use of the 3D production engine "Blender." Such a course extends historical thinking beyond mere data collection, organization, and analysis, providing a foundation for cultivating intuition and insights that enable data interpretation within the realm of historical research [27].

The course was conducted on desktop computers running Windows 10 with Blender installed in a computer laboratory. The course has two primary objectives. First, it aims to encourage learners to observe and enhance their digital data literacy by enabling them to independently create and witness the realization of imagination and creativity from data. Second, it aims to reduce entry barriers for humanities major students who are non-specialists in the digital and data fields by allowing them to personally experience the effectiveness of EdTech and enter the perceived complex realms of digital and data. Most students who participated in this course were majoring in history, history education, Korean literature, Buddhist studies, and law. Via pre-course interviews, it was discovered that they had a keen interest in the digital data field but were unsure about how to initiate their engagement. This course involves guiding them to analyze the data they wish to implement, teaching Blender manipulation skills, and allowing them to "feel" their imagination tangibly as a concrete form via EdTech education, with the intention of shifting the paradigm of education.

This course consisted of a total of 15 sessions, divided into two components: 30% domain knowledge lectures for extracting and analyzing data in the first and second sessions and 70% 3D Blender modeling lectures. The course title, "3D Time Machine," was created to evoke the idea of collecting past data and implementing it in 3D to recreate one's current environment. It is a software-centered EdTech. Just as historical accuracy is required to implement historical elements, learners accumulate domain knowledge about the subject they have chosen, thereby increasing their data literacy. As "3D Time Machine" is not just a simple 3D creative design course, instructors must possess the requisite expertise to provide guidance regarding the intended implementation. The process of researching and analyzing materials related to the chosen subject and building domain knowledge continues throughout the course, contributing to learners' increased data literacy [28]. Digital literacy is enhanced via the practice and manipulation of Blender techniques until the chosen subject can be realized using one's creativity. This is the essential form that metaverse and futuristic skills education courses in universities should take. To transform imaginings into virtual reality, one must understand the subject matter (data literacy) and learn the implementation techniques to visualize one's ideas (digital literacy) [29]. The successful results achieved in the "3D Time Machine" course can be utilized as the backdrop for metaverse platforms or as metaverse content.

### 2.2. Blender Course Curriculum and its Implementation

The "3D Time Machine" course focused on enhancing digital literacy via Blender, covering (1) "3D shape manipulation," (2) "color manipulation," (3) "rendering," and (4) "add-ons," with an emphasis on mastering these functions via practical application. The evaluation of this course is based on two main components: (1) projects that adequately incorporate historical source data–for this purpose, data extracted from historical sources should be convincingly integrated into the assignments; and (2) projects that demonstrate proficiency in Blender's fundamental techniques. This course spanned a single academic semester and comprised a total of 15 sessions. The basic structure of the course is outlined in Table 1 below:

**Table 1.** Curriculum of "3D Time Machine" for 15 weeks.

| Lectures | Session | Contents |
| --- | --- | --- |
| Data Literacy | 1 | Data extracts |
| | 2 | Data analysis |
| | 3 | Domain knowledge study |
| Understanding Blender Basics | 4 | Purpose of learning Blender |
| | 5 | Principle of Blender |
| | 6 | Interface of Blender |
| Proficiency in Blender Manipulation | 7 | Object making, Add-ons |
| | 8 | Modifiers, Lighting |
| | 9 | Blender nodes, Blender Kit |
| | 10 | Rendering |
| Project | 11 | Individual Practice for Project 1 |
| | 12 | Individual Practice for Project 2 |
| | 13 | Individual Practice for Project 3 |
| | 14 | Individual Practice for Project 4 |
| Presentation | 15 | Presentation in Public |

Teaching Blender manipulation alone from the first to the third session would be tantamount to providing a private institution course rather than a common university

lecture. These initial three sessions were dedicated to developing data literacy in the context of the "3D Time Machine" course. Sessions four through ten concentrated on instructing Blender operations. They formed the segment of the course dedicated to advancing digital literacy within the context of the "3D Time Machine" course. During sessions four to ten, students first learned the principles and structure of the 3D design engine, and subsequently, they participated in sessions seven to ten, wherein the instructor progressively created a single piece of content using Blender while students followed along.

To facilitate hands-on learning, the researcher created content that allows students to use a Shilla era sword within the metaverse. In preparation for realizing an image of a sword that might have been used in the Shilla era, various related data, such as those shown in Figure 3, were collected and studied. Subsequently, students utilized this data to create Figure 4, integrating historical imagination. In software-driven EdTech environments, these opportunities where instructors demonstrate and students follow are highly valuable.

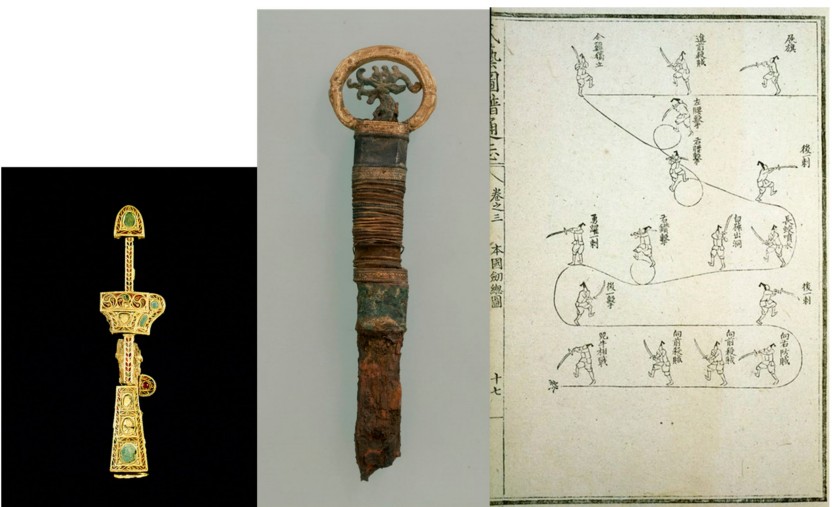

**Figure 3.** Pictures of Shilla swords and a drawing of martial arts.

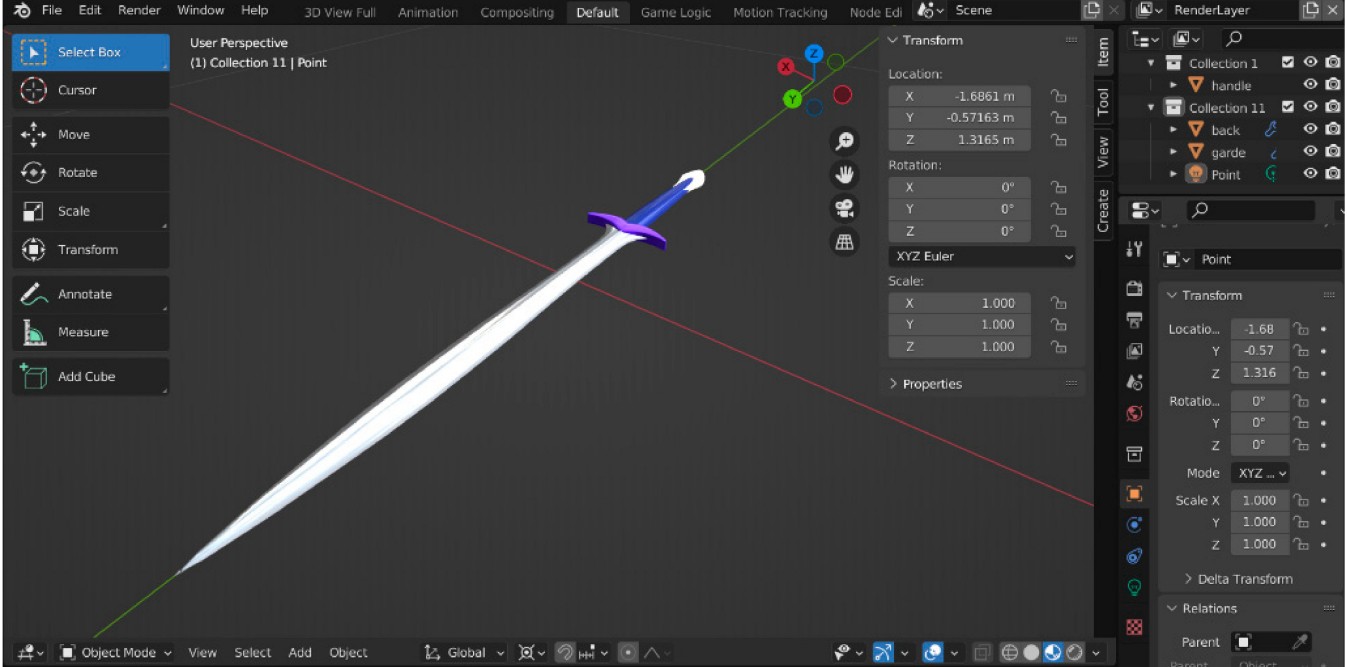

**Figure 4.** Blender interface and the process of implementing a 3D Shilla sword.

Furthermore, these implemented results were exported as "fbx files," as illustrated in Figure 4, allowing them to be integrated into metaverse platforms for use, as shown in Figure 5. The process of exporting 3D modeling outcomes to different software and adapting them to the platform is the aspect where students often encounter the greatest challenges. Unlike hardware-centered EdTech, software-centered EdTech involves the use of various software applications, necessitating a solid skill set in operating each of these applications.

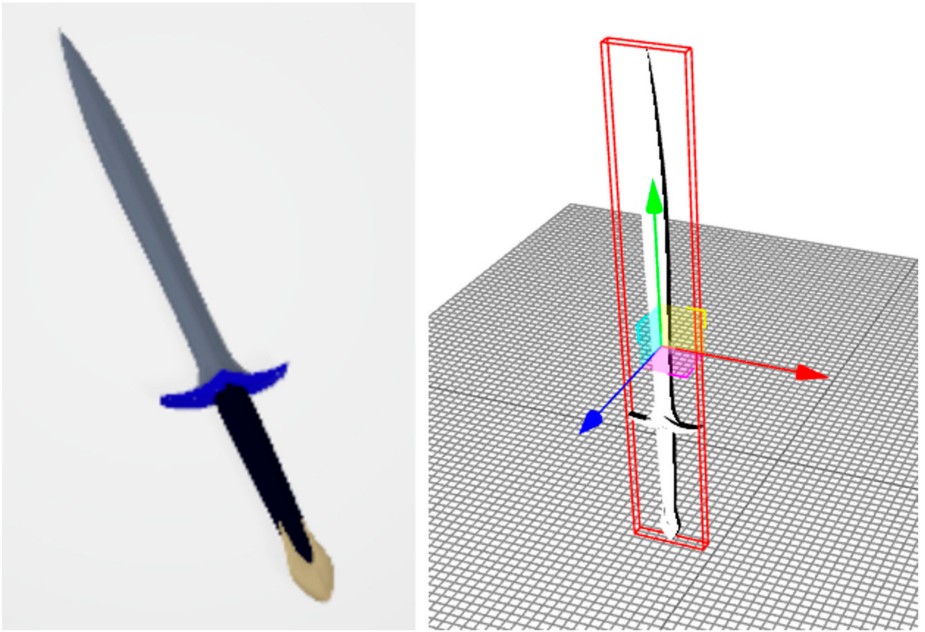

**Figure 5.** "fbx file" image of Silla sword and its AR image.

From the 11th to the 14th session, students engaged in a mini project where they used Blender to implement the research topics they had chosen. They were encouraged to select relatively lesser-known subjects for their research. This recommendation stemmed from the idea that well-established research subjects might hinder creative thinking and the adoption of new ideas, as they could subconsciously steer students toward preconceived prototypes. For example, a student majoring in Korean literature with a deep interest in historical and literary imagination chose Chunhyang's room as the research subject. This was a topic rich in both literary and historical imagination. Another student from the Korean literature department selected the smithy of the Joseon period, focusing not only on the physical space but also on recreating the various ironworking tools from that era.

In contrast, history and history education students chose subjects that were historically verifiable but not widely recognized, such as Sosheowon, Wanwigak, Hyukgeose's Najeong, Andong Byeolgung, Pungpaegijigwan, and Ujeo seowon. Blender offers numerous versatile features for creating virtual spaces, making it an excellent tool for implementing these historically significant places. Furthermore, students also considered relatively well-known but internally unexplored places like the Russian Legation, Jangyongyeong, and Seodaemun Bus Stop as their research subjects.

In software-centered EdTech, it is essential to recognize that once students acquire the technical skills, they gradually come to realize how to tailor "what" and "how" they create to match those skills. Such realizations manifest themselves in the educational process that crystallizes imaginative activities within a realistic virtual context. Some of the representative research subjects that students individually selected are as follows:

1. Smithy of Joseon era;
2. Soseokwon;
3. Wanwigak;

4. Chunhyang's Room;
5. Hyukgeose's Najeong;
6. Jangyongyeong;
7. Russian Legation;
8. Seodaemun Bus Stop;
9. Sontag Hotel;
10. Andong Byeolgung;
11. Shanghai Provisional Government Building;
12. Pungpaegjigwan;
13. Woojeo Seowon.

To enhance students' data literacy, each student was encouraged to read a minimum of two relevant research papers related to their project research topic. The researcher also initiated investigations on each topic. The "3D Time Machine" course extends beyond being a mere 3D sculpting and design class. Therefore, the instructor must possess sufficient knowledge to guide students about their research topics effectively. In other words, it encompasses a creative procedure wherein students employ their internalized and personally established historical data, rather than solely replicating and employing pre-existing results. This entails substantial engagement in communication between the instructor and the learners, propelled by interactions based on data.

*2.3. Results of the "3D Time Machine" Course*

One of the most significant outcomes of the "3D Time Machine" course is the enhanced communication between the instructor and the students. Throughout the course, students demonstrated a remarkable increase in their concentration, rigorous preparation, and, notably, an increase in their verbal participation. With a strengthened domain knowledge of their chosen project topics, they exhibited enthusiastic confidence in articulating how they would implement 3D designs. Furthermore, unexpected and creative questions regarding the utilization of Blender poured in, intensifying the overall level of interaction between students and the instructor. For example, in the case of the student majoring in Korean literature who chose to recreate "Chunhyang's Room" as their project, the following represents a portion of the domain-specific data, "Chunhyangga," that they engaged with:

In the room, they brought her in, and upon looking around, they found the room to be exquisitely adorned. The walls were painted with meticulous care, and the wooden floor was pristine. The wall coverings, the decorations, and the ceiling were all adorned with ornate designs. On the highest-quality bureau, there was a stationery set, and next to it lay some books. At the front, there were incense burners, and a study desk was placed. A large mirror was attached to the wall above, and it served as a dressing table. There were essentials like a washbasin, a brush stand, a tobacco cabinet, and an ashtray, all placed in the middle of the room. The walls were adorned with paintings of young ladies, and nothing else was affixed.

The painting on the east wall depicted a night rain on the Soyang River and a moon rising over the Dongjeong Lake. The radiant moonlight shone in the deep forest where two ladies in white attire rode on horseback. They appeared to be dancing, and there were also a mirror and a comb. On the south wall, there was a painting where an enemy ambushed all around under the autumn moon near Gwisan Mountain. Amid the boisterous enemy camp, King Cho, known far and wide, was drinking wine inside his tent. A beautiful woman, dressed as a man and carrying a sword, was approaching, poised to thrust her dagger into his neck.

When looking towards the west wall, there was a garden blooming with flowers at Changshingung Palace, and the rough grass filled the yard. A passing crow cast a shadow towards Soyangjeon, while a lady with a silk fan was gazing. As for the north wall, it portrayed the joyful scene in Geumgok, a banquet with an unexpected calamity in a matter of moments. In the forefront, soldiers in armor were gathered, their eyes as sharp as needles. It was a critical moment, and the four directions were in peril. Beautiful lady's gazed upon

it, an exquisite woman with a silk fan, and the scene looked as if she were a falling blossom in the Pavilion of Spring Breezes during a tranquil March.

Inevitably, they could not disagree and walked into the room together, holding each other's hands. Upon entering, they noticed that all four walls were adorned with paintings. These depicted various scenes, including Buchoonsan's Eomjarung and the bustling market at Gangidaebu. Qin dynasty's talented lady Doyeonmyeong was depicted, along with scenes of a shaky little boat sailing to Semyang.

Additionally, Chaeseok river showed Taebaek Lee enjoying grape wine, attempting to catch the moon beneath the water, eventually sitting on a whale's back to ascend to the heavens. There was also a depiction of four elderly individuals beneath a pine tree gazebo at Sangsan. One elderly man held white stones and sat quietly, while another clutched black stones and did the same. Beside them, another elderly person, seemingly engaged in a lively discussion, held a chessboard. Under the pine tree pavilion, another elderly person was pictured in a merry state after drinking Cheonilju. There was also a depiction of a little table adorned with a tea set, a small kettle, and an incense burner with smoke gently rising, alongside the splendid cabinet, the masterwork calligraphy table, and the charming inkstone placed upon it [30].

From this perspective, it is apparent that within the framework of the digital data narrative course aimed at augmenting students' digital data literacy, EdTech functions not only as a tool to enhance academic proficiency and skill development for future employability but also as a facilitator in directing lectures and fostering academic communication between instructors and students. The following Figure 6 represents some of the mini projects created by students in the "3D Time Machine" course.

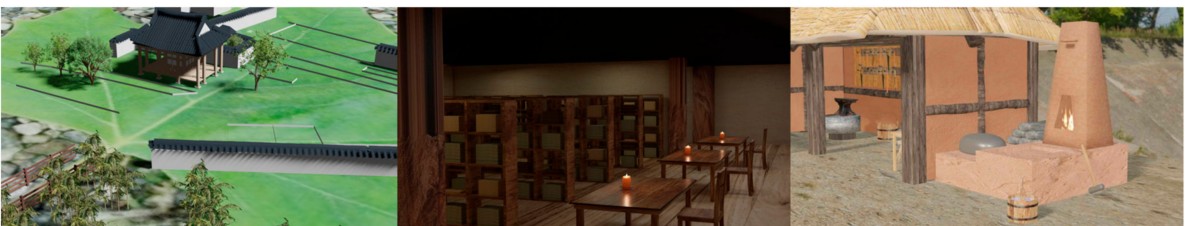

**Figure 6.** Sosheowon, Wanwigak, and Smithy of Joseon era.

Among these projects, a student who majored in history and had an interest in landscaping entered the "3D Time Machine" course. This student wanted to understand the trees in the gardens of the Joseon era, the trees used in contemporary landscaping, and the changes in landscaping methods based on different eras in the Korean Peninsula (Unified Shilla, Goryeo, and Joseon era), as well as the differences in landscaping between East Asian countries (Korea, Japan, and China). They aimed to visualize a "Joseon era garden in 3D."

To build domain knowledge, both the student and the researcher collected and analyzed data on various gardens, such as Seoseokji in Yeongyang, Dasanchodang in Gangjin, and Choganjeong in Yechon. They went beyond collecting photos or illustrations conducting thorough investigations that included site plans and details of plantings, among other data. They also reviewed academic papers. After exploring several gardens, this student drew inspiration from Figure 7, Sosheowon, to initiate their mini project. They made considerable efforts to match the seeds and colors of trees in the garden based on domain knowledge. They even learned to create 2D/3D design plans using "Auto CAD" and "CAD" software for drafting and brought the site plan file into Blender for use. They studied the details of the garden's features. To obtain specific data, they read papers such as "A Study on the Structure of Sosheowon Landscape Garden Featuring Borrowed Scenery: Focusing on the Sosheowon Sisun and the Thirty Poems of Sosheowon" [31] and "A Study on the Restoration of Traditional Garden Spaces for Garden Restoration: The 40th Mingseung: Damyang Sosheowon" [32] and "Study on the restoration of Sosheowon Garden's

Goam-Jeongsa and Boohwondang buildings." [33] These papers contained abundant data to evoke Sosheowon. This is what narrative means in education. It empowers learners to take charge from selecting their subject to producing results.

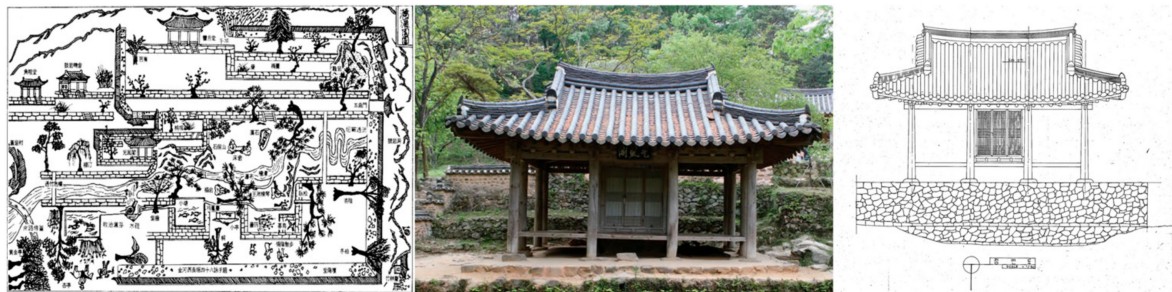

**Figure 7.** Drawing of Sosheowon and Gwangpoonggak with drawings.

And the outcome, formed by combining diligently acquired domain knowledge over a semester and refined Blender skills, is depicted in Figure 8 below.

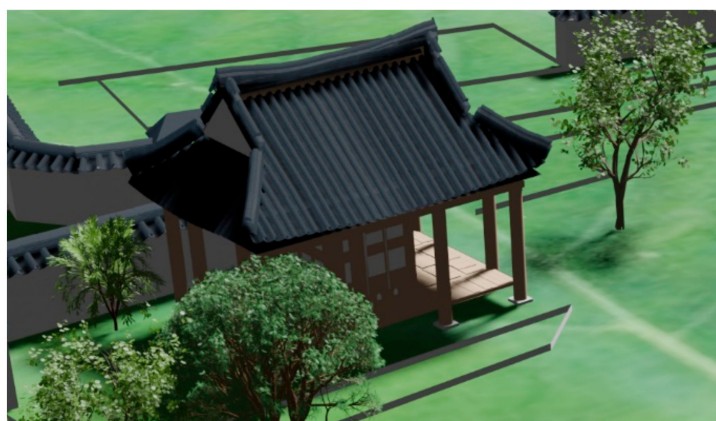

**Figure 8.** Gwangpoonggak restored by Blender.

The students exhibited considerable interest in recreating traditional Korean architecture, and the quality of their work, considering one semester of study, was notably high. Below is Figure 9 illustrating the 3D modeling implementation of Andong Byeolgung.

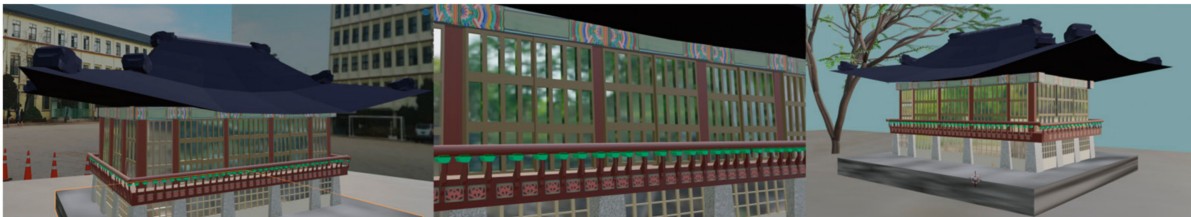

**Figure 9.** Andong Byeolgung is restored by Blender.

The successfully implemented results can be utilized both as a background in metaverse platforms and as content within these platforms, as demonstrated in the images below. Figure 10 illustrates an actual building of the exhibition hall and its metaverse counterpart at Jeonju Craft Exhibition Hall. Figure 11 presents an image from the global metaverse game platform Roblox, which was created using historical data from the missionary work in Seongyojang of Gangneung. All these outputs bear a resemblance to the results of the "3D Time Machine" course. Hence, in the metaverse platform market,

such methods of implementing historical data via "3D Time Machine" are indeed being practically employed.

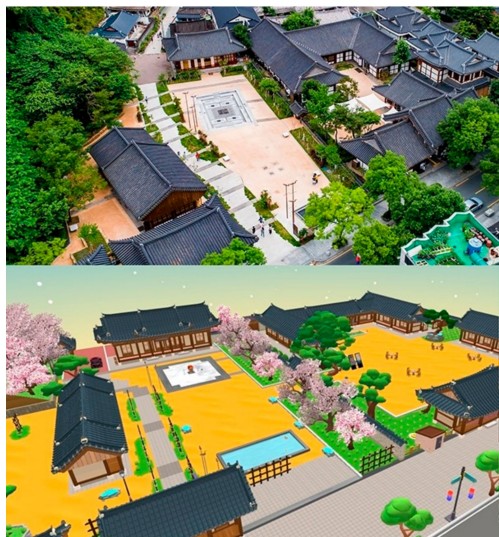

**Figure 10.** Jeonju Craft Exhibition Hall and the metaverse image.

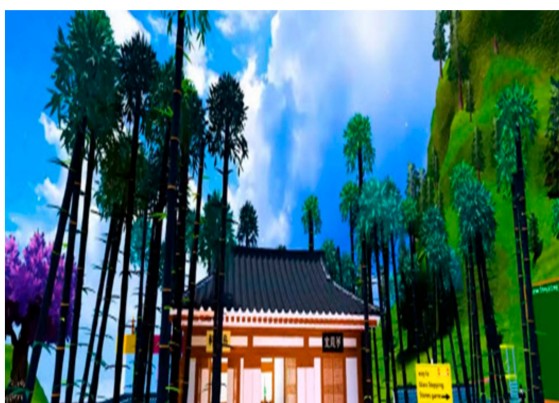

**Figure 11.** Seongyojang 3D Image in Roblox.

## 3. Student Interviews Logs

Student A: This semester, I was contemplating which courses to take, and I couldn't help but notice the unique educational approach that this program offered, which was quite distinct from the conventional courses in history. I vividly remember something you mentioned when I first arrived. The professor said that by learning computer skills in addition to history, we could compete effectively with students from other departments, and this statement left a lasting impression on me. Taking this course, I initially found 3D modeling to be incredibly fascinating through the lectures and YouTube tutorials. Consequently, I dedicated more time to practicing and improving my skills. It felt like I was engrossed in something I hadn't done for a long time. I am sincerely grateful for the opportunity you provided. If you continue to offer courses in the 3D domain next semester, I would definitely like to enroll.

Student B: At first, I was a bit perplexed about the course content and even questioned if this was something that could be taught in the liberal arts, given my background. However, as I delved into the course, I realized that these preconceived notions were limiting. I came to understand that there are more possibilities for liberal arts and history majors than I had thought. Although it was quite challenging initially, and still feels challenging to be

honest, the sense of accomplishment and enjoyment from creating and learning outweighs the difficulty. I genuinely look forward to learning more in the coming semester.

Student C: Initially, I decided to take this course with the aim of obtaining a "Micro Degree," and there was a sense of anxiety, wondering, 'Can I do well in this?' as I had never attempted such programs before. However, learning 3D modeling and the process of creating something from one's own ideas felt incredibly rewarding. In addition to making objects for assignments, I started creating things like doughnuts and building shapes for personal skill improvement, which sparked my interest in advanced Blender techniques. Nowadays, 3D is widely used in various broad and public domains like museum promotion and exhibition design, so I believe that the time spent learning Blender can become an asset in the future. Moreover, your passionate and kind teaching made the class enjoyable. I used to think that I lacked talent in anything related to computers, such as 3D and programming. It was surprising to discover that I found joy in learning. As I learned, I realized that I could create things I had envisioned even better. This experience gave me the confidence to approach any other 3D program in the future.

Student D: Since starting my studies, I haven't taken many computer-related courses. I've only taken three courses so far: Computing Thinking, Digital Forensics Fundamentals for my major, and this semester's "3D Time Machine." My initial experience with Computing Thinking wasn't great, as I received the worst grade possible, and while I did well in Digital Forensics Fundamentals, it was quite exhausting. When I enrolled in the "3D Time Machine," I was worried that it would be as challenging as those two previous courses. And indeed, it was difficult. I had trouble keeping up, and without your guidance, I often had no idea what I was doing. Nonetheless, despite the challenges, I found it enjoyable. Handling a program I'd never used before was refreshing, and, most importantly, your continuous support and guidance played a significant role in helping me finish the course on a positive note. I think your ability to create a positive atmosphere in the classroom allowed me to trust and follow your lead. I was gradually losing interest in my major courses in history, as they were mostly writing-focused and often felt monotonous due to repetitive content. This course, however, was engaging. It was the most interesting course I've taken in history. History courses have always focused on writing, and they often felt repetitive and dull. This course was different. I believe this course is worth trying for fellow students aspiring to work in history-related fields.

Student E: When this course was first offered, I hesitated about whether to take it, and in the end, I didn't enroll, partly due to scheduling conflicts. However, last spring, I observed my peers taking this course and doing assignments using 3D modeling in other courses. It seemed impressive and intriguing, and I wanted to try it myself. So, I decided to enroll this semester. Initially, I found every terminology challenging, but I still enjoyed learning something new. A small cube became a miniature house, and later it turned into more massive structures. One shape transformed into flowers and even a boat. As I reflect, I think that's how technological progress likely happens, through small steps. Even now, while writing this feedback, I find hotkeys confusing, and sometimes the program doesn't behave as expected. I also remember the peculiar behaviors of the program at my workstation. Nevertheless, I'm grateful for showing me how fascinating and enjoyable it can be to witness a small cube turning into a building! To survive in the future, continuous improvement is essential. Therefore, I won't forget Blender and will keep practicing. I'm sure you are working hard, and if you ever need assistance in the future, please don't hesitate to call on me. Thank you for this semester!

Student F: Regarding the general content of the course, I had heard about it from seniors, but I had no idea about the specific programs or the course structure. Therefore, I initially enrolled in the course with the thought of learning a computer program that could be integrated into history. Also, while I had heard the terms "3D program" and "metaverse" quite often over the past few years, I felt that they were far from my reach, and I believed that the technology would be challenging to learn. However, during the course, I realized that Blender, the program we used, had a high level of accessibility, and beginners could

learn it easily. As someone who enjoys hands-on activities, regardless of the quality of the final product, I found it exciting and fascinating to create something with a computer, something I'm already familiar with. The course provided not only the opportunity to learn the program but also insights into the practical applications of these technologies, which I vaguely knew were used in my favorite games and movies. Knowing the specific processes involved in creating characters or backgrounds was also interesting. In my third year, when there were no light electives available, I found it challenging to get through the semester, but I enjoyed participating in this course without any burden, as it allowed me to engage with computers and learn new things. Personally, I was highly satisfied with this course, to the extent that I would recommend it to my close juniors who have many days left at school. Of course, this was only possible due to the professor's efforts to create an environment where students could easily ask questions and his enjoyable teaching style. I hope to have the opportunity to take another course with him next semester.

Student G: Before enrolling in this course, I had heard about it in passing, but I had no clear understanding of it. During the course, as we used the Blender program, I became curious about creating something in "3D." However, I got somewhat confused with the wealth of information. If it were a program like Power Point or Word, which I use daily, I would have had a rough idea of where certain functions are located. This could have made it easier to remember the course content. But since this was a program I was encountering for the first time, while I tried to follow along during lectures, it was a bit challenging to recall everything later. Nevertheless, through repetitive use of specific functions every week, I was able to use them in assignments, despite my lack of expertise. Additionally, because there was little pressure from assignments and exams, I could learn comfortably without haste. Lastly, taking the "3D Time Machine" course was a valuable experience in broadening my horizons. During the course, using Blender, I gained insights into how my major field could expand, and during my visit to the expo, I had a chance to witness that reality firsthand. If someone were to inquire about the "3D Time Machine" course, regardless of their prior enrollment in other non-major courses, I would recommend it as a must-take course at least once.

Student H: The "3D Time Machine" course was the best choice of my last semester! I had seen my history major friends enjoying this course, and I had thought of taking it someday, but luckily, I could take it this semester. Frankly, this course is known for its heavy workload and difficulty, largely because it overlaps with my law and history major courses. My friends kept persuading me to switch to another course, but after attending the professor's orientation and the first few hours of lectures, I was determined to stay. Despite the stress from my graduation thesis and assignments, this Thursday class was a significant source of relaxation for me. Being able to apply what we learned and create results immediately was enjoyable in itself. Moreover, thanks to the professor's consistent encouragement, I could confidently utilize Blender. I briefly contemplated reconsidering my career choice, but after seeing my humble assignments compared to the works at the expo, I regained my focus. Nevertheless, I had a great time at the expo. Seeing how the metaverse can be utilized, especially in creating cultural heritage and historical content, was delightful. It made me realize that to achieve such results in 3D, it requires a mountain of learning, which motivated me to strive for better outcomes. I worked hard on my laptop for several days, but I still find it embarrassing to show my work to an expert like the professor. I'm sure if I had mentioned my topic beforehand, you would have discouraged me. Still, I wanted to create something meaningful and gave my best, though it was challenging. The royal palace was tough. I would like to try making animations with Blender next semester, but it's a pity that I'm graduating. I hope the professor continues to teach here, so more students can experience this delightful course. I'm truly grateful for this semester. Hope to see you again someday.

Student I: As a student in the College of Liberal Arts, I couldn't participate in courses that involved programming or creating something with specific software, due to my law and history major requirements. However, this course allowed me to combine the historical

knowledge I possessed with practical skills, resulting in tangible creations. Although some aspects of this hands-on work were tedious, I felt a great sense of accomplishment. This led me to contemplate taking a course during my final semester where I could combine the knowledge and specific skills I had acquired over the years to produce something. Initially, it was a bit challenging to adapt to the course. While I had worked with a computer on a two-dimensional plane, much like drawing on a canvas, tasks that required using the x, y, and z-axes in a three-dimensional space were new to me, which made things a bit difficult in the beginning. As the course progressed, I also faced some challenges due to issues with my computer. Sometimes, my screen would go black while working on demanding tasks, causing me to lose my progress, which was frustrating. Nevertheless, I believe this course expanded my horizons and changed how I view the world. Before, I had assumed that the only way to apply my knowledge in history to society was through careers like a history teacher or a researcher. However, this course taught me that there are various ways to integrate historical knowledge into different fields, beyond academic research. Although I'm preparing to enter the legal profession, and there might not be opportunities to combine my legal studies with history, I've become interested in learning more about Blender as a hobby if the chance arises in the future. I appreciate this enriching course over the semester.

## 4. Analysis of Written Statements and Interviews

### 4.1. Content Analysis

From the opinions submitted in writing by the students, "adjectives," "nouns," and "verbs" were extracted and categorized for analysis. The findings indicated a notably positive perception among students toward the "3D Time Machine" course, which utilizes 3D modeling. They regarded it as a departure from the conventional educational paradigm, expressing both the challenging nature and fatigue associated with software-centered EdTech. However, they also acknowledged its relevance to their professions and its impact on their thinking patterns as shown in Figure 12. Specifically, they highlighted experiencing expanded cognitive abilities and perspectives, impacting both their professional and cognitive dimensions positively.

This underscores the necessity of targeting individuals preparing to navigate a digital society for substantial digital education that fosters sustainability skills. Additionally, a crucial point gleaned from the students' feedback was their enjoyment of the course. This enjoyment, closely linked to a sense of achievement, reflected their "active participation in history" rather than mere observation from the sidelines, finding pleasure in creating something tangible within history and witnessing the outcomes firsthand.

### 4.2. Thematic Analysis

For thematic analysis, patterns were classified into (i) "future job before the course" (Figure 13)–"future job after the course" (Figure 14), (ii) "what happy" (Figure 15)–"what sad" (Figure 16), and (iii) "what good" (Figure 17)–"what bad" (Figure 18). (i) pertains to responses to the questions before and after the course, (ii) relates to opinions expressed during the course, while (iii) involves responses emerging after the course.

The most noticeable change observed among students after the class primarily resides in their perspectives regarding future careers. While many students initially contemplated becoming history teachers before the course as shown in Figure 13, Figure 14 tells that there was a notable increase in interest in professions related to digital fields such as 3D modeler, contents developer, digital designer after the course.

The captivating aspect of the "3D Time Machine" course lay in the direct act of "creation" it offered to students (purple in Figure 15). They accomplished results that projected historical imagination (pink in Figure 15). It demonstrates that education was not solely focused on functionalities but that students' thinking was influenced by digital means, which were then realized once again via digital platforms. As students became adept in digital environments, they found joy in creating their own thoughts. This aligns

precisely with the philosophy of the digital data narrative curriculum advocated by the author, unfolding one's data in a manner akin to writing. Additionally, it aligns with the educational psychology of EdTech.

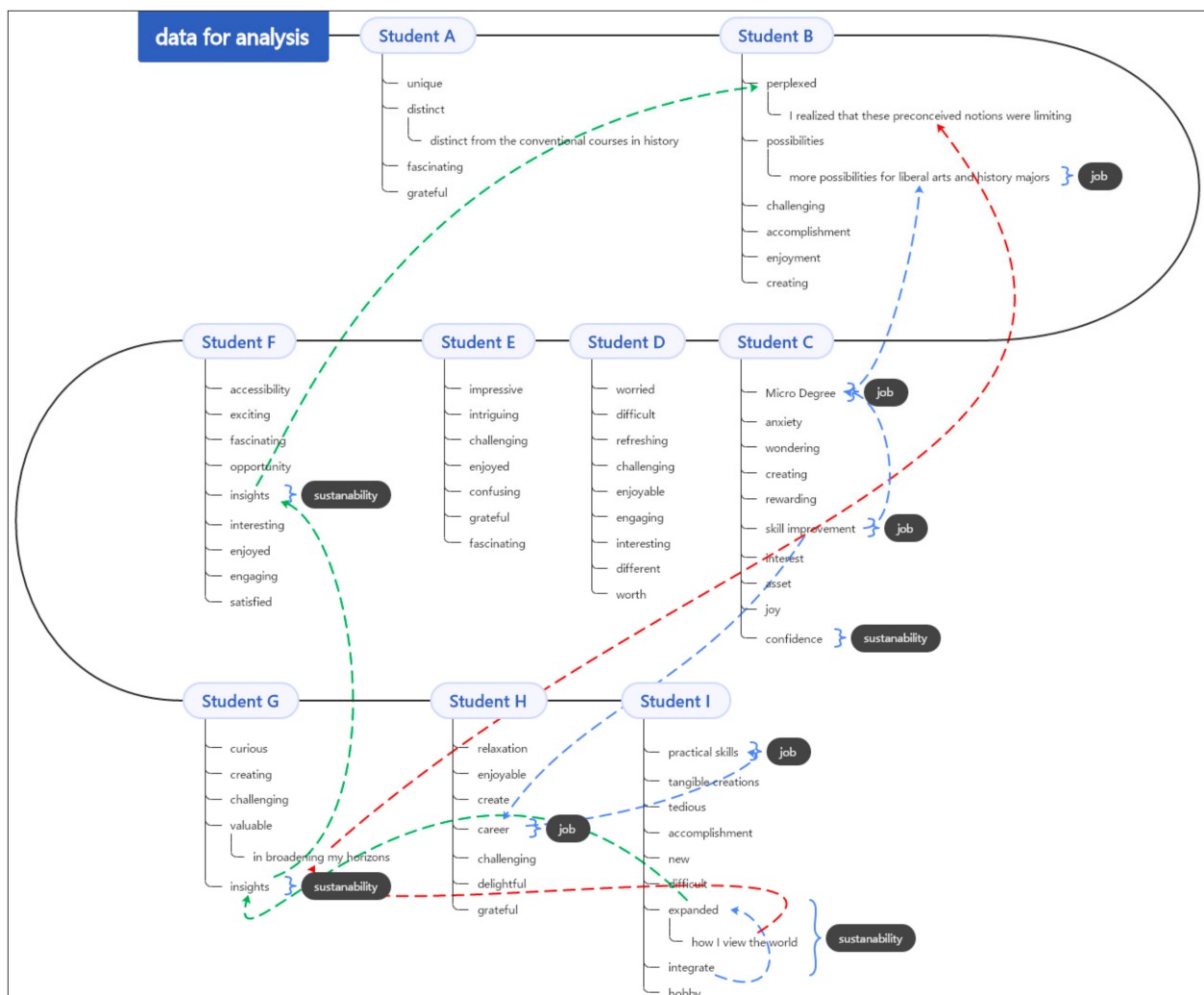

**Figure 12.** Data analysis of the "3D Time Machine" course's student interviews.

The students acknowledged that the course was not that easy, refraining from describing it as "difficult" but rather as "challenging." This reflects the need to teach digital skills systematically and diligently. Those who responded that the class was "complex" (blue in Figure 16) felt the complexity not just in the intricacies of digital functionalities but also in the complexities associated with translating historical data into digital formats.

The reasons behind the favorable perception of the course in Figure 17 were also linked to careers (yellow in Figure 17). Some respondents indicated gaining confidence (blue in Figure 17) as an intangible asset and learning to merge analog with digital realms. It is crucial to seriously consider responses where the class became a pivotal point in their thinking (purple and pink in Figure 15). This is because software-centered EdTech should take center stage in digital education, not solely for the purpose of cultivating digital functionalities.

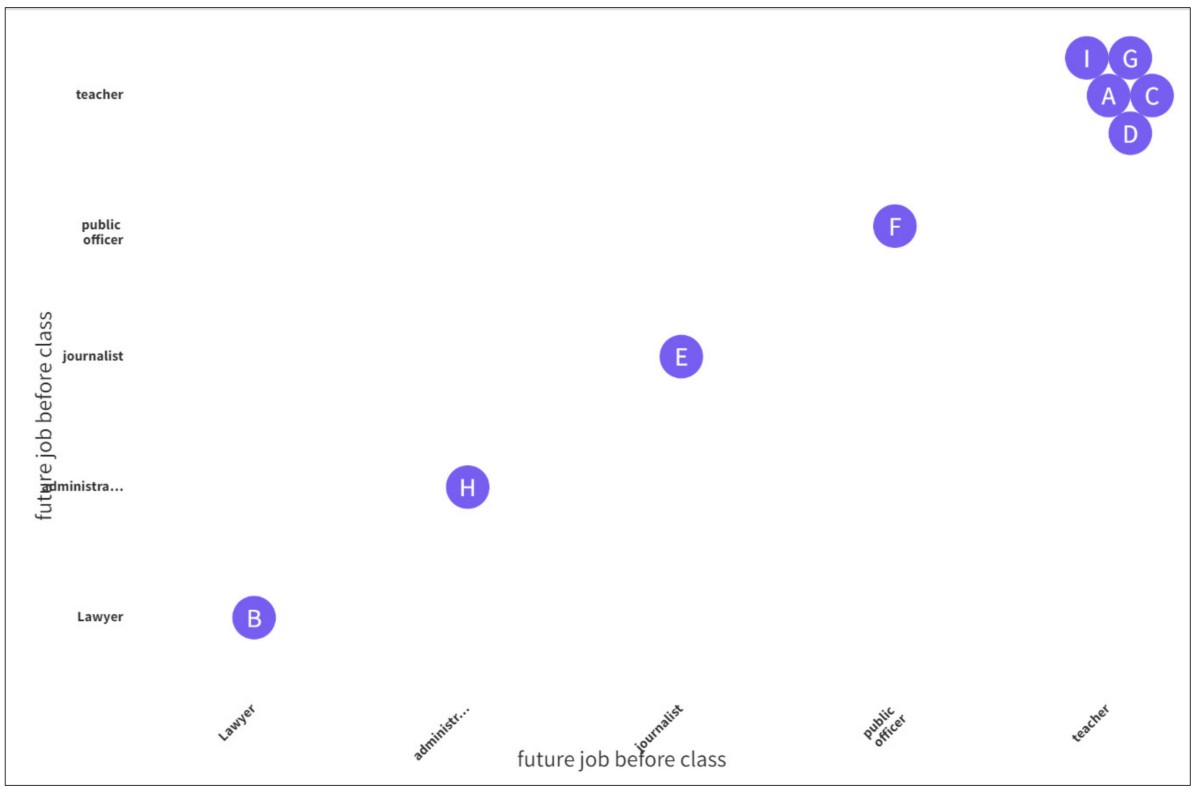

**Figure 13.** Future job interview results before the course.

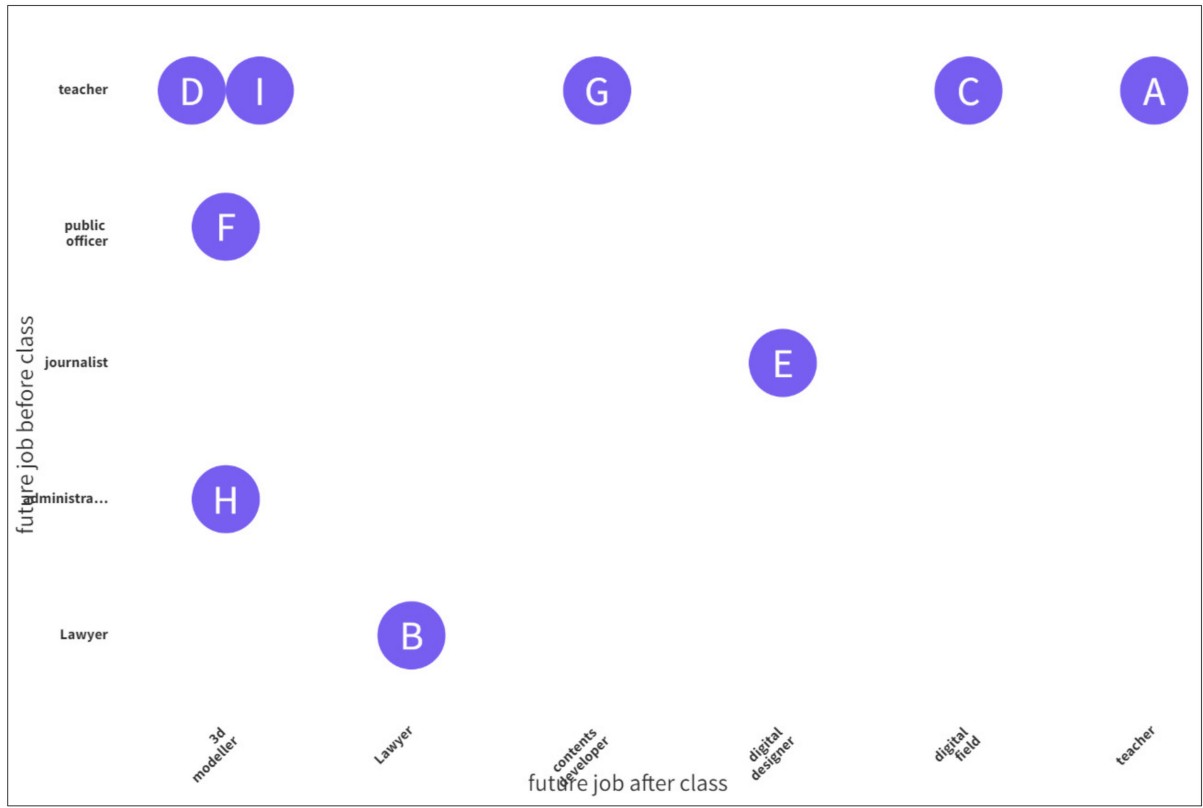

**Figure 14.** Future job interview results after the course.

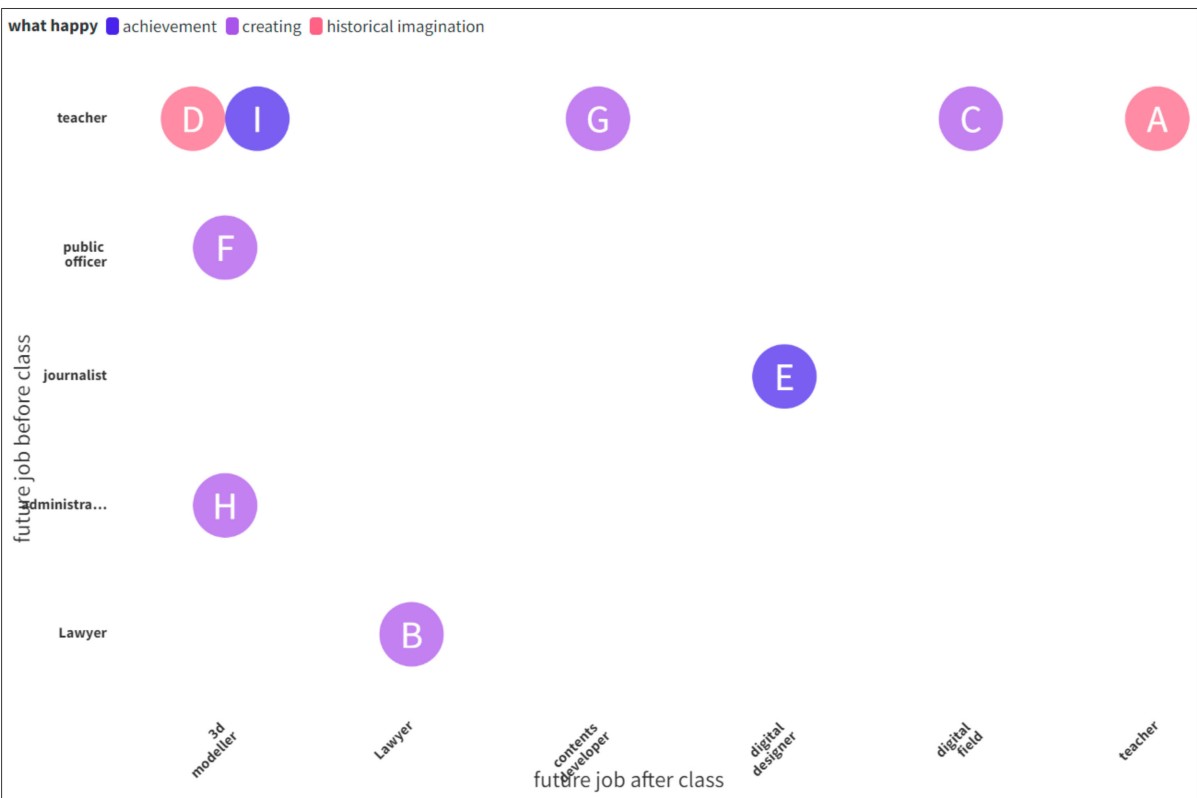

**Figure 15.** "what happy" interview results.

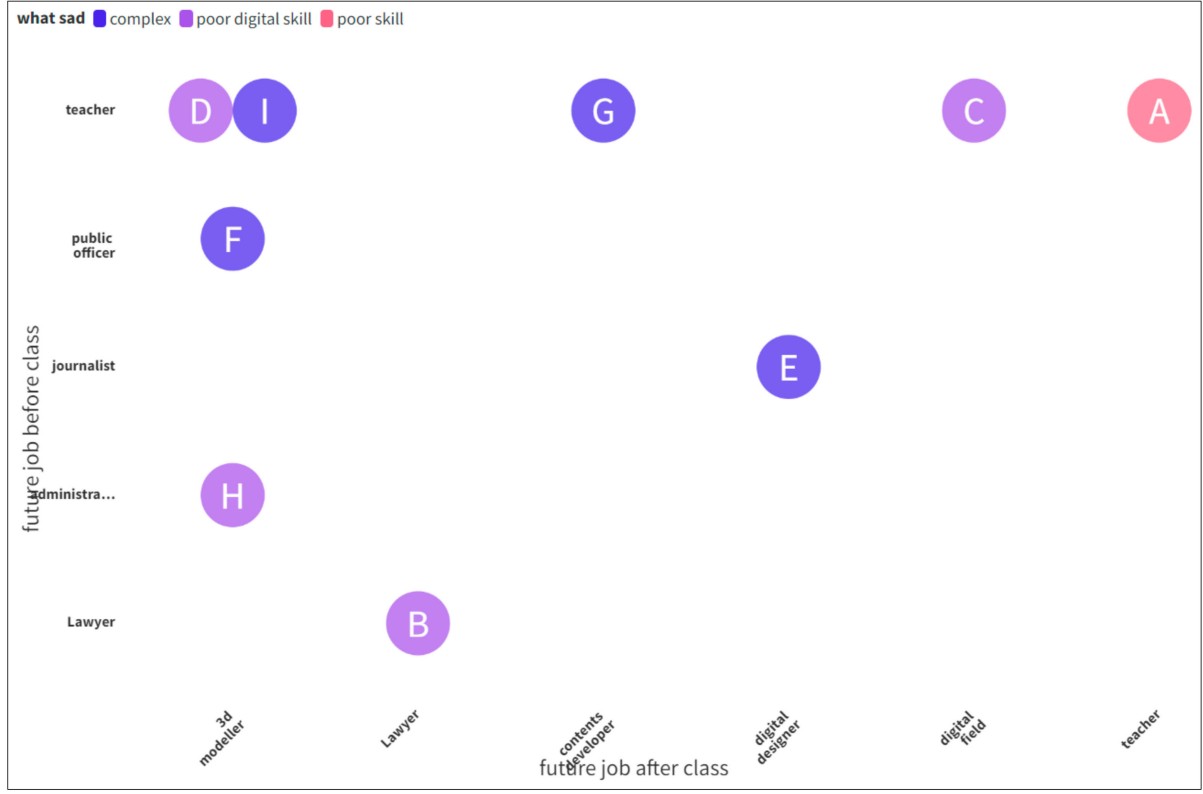

**Figure 16.** "what sad" interview results.

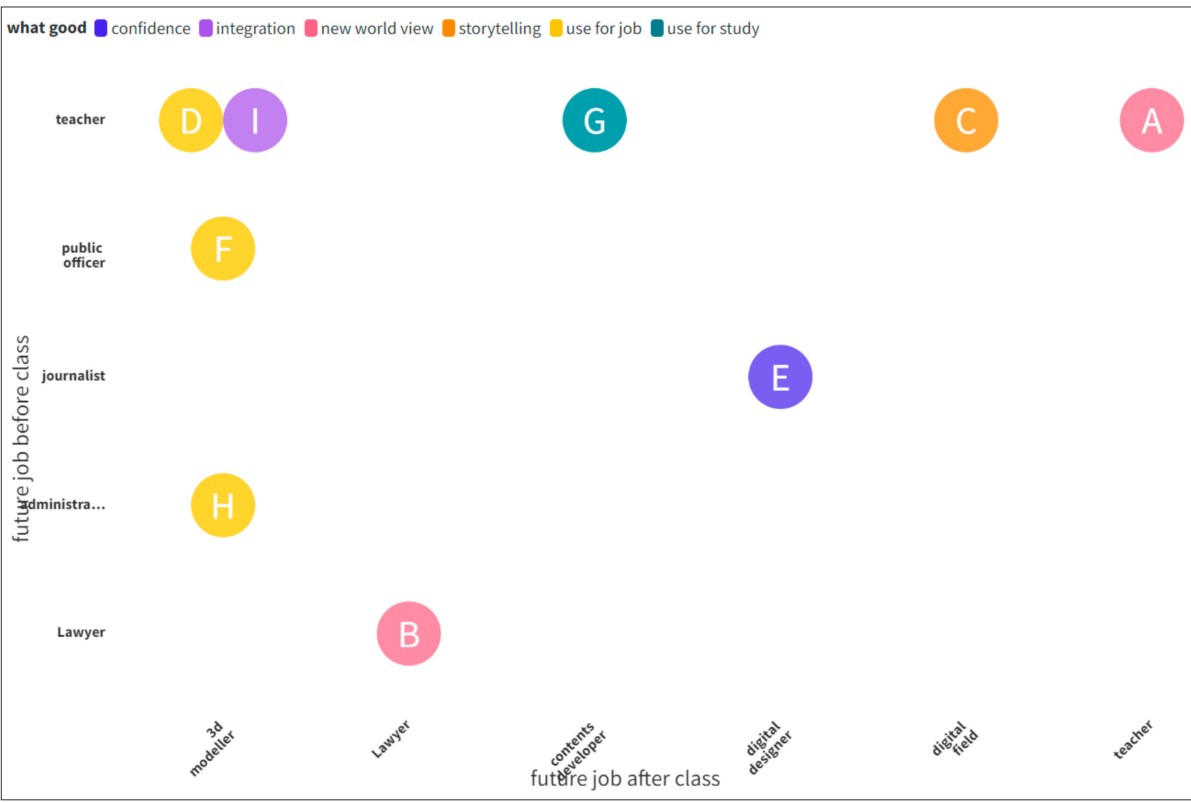

**Figure 17.** "what good" interview results.

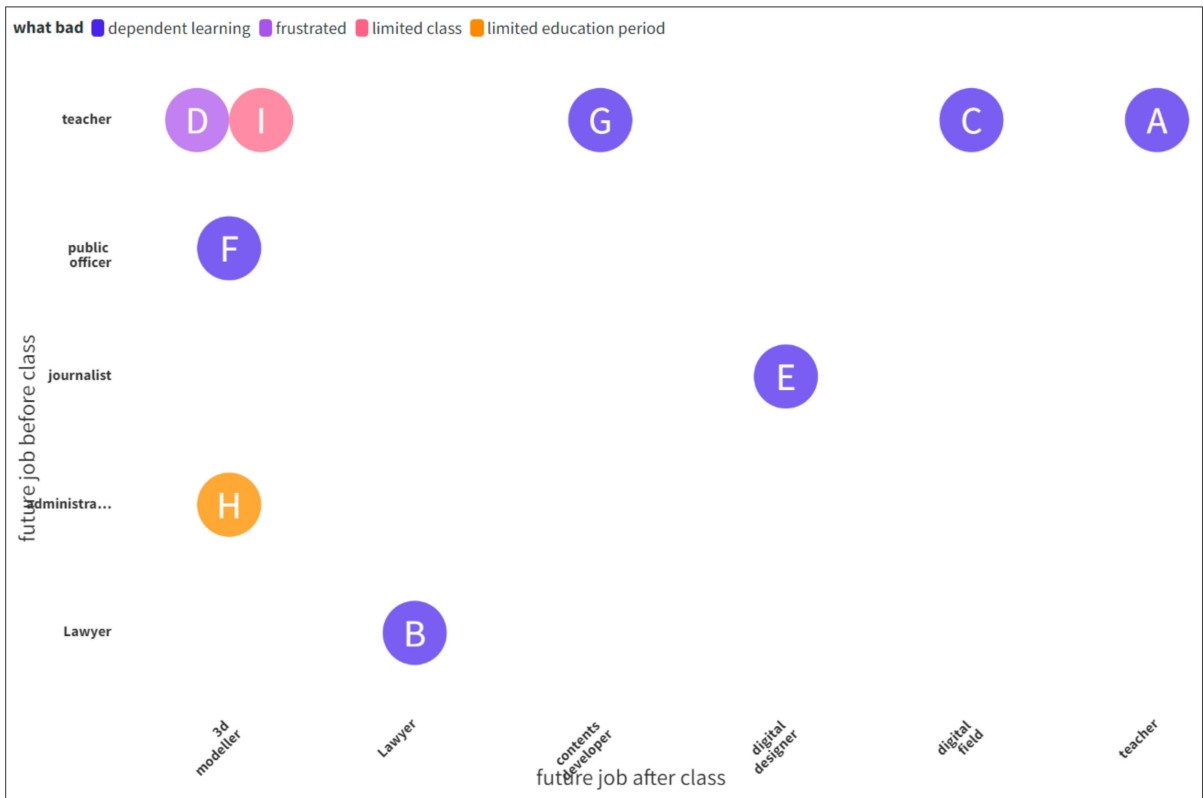

**Figure 18.** "what bad" interview results.

Most students expressed regret about not being able to continue studying on their own after the course (blue in Figure 18). This relates to the limited duration of the course (pink in Figure 15). With only 15 sessions in a semester, the scope for achieving educational objectives across the elementary, middle, and high school curricula is limited. We also find this aspect to be the most regrettable. Among the students' responses about "what is regrettable after the class," the term "vagueness" often appeared, indicating an emotion stemming from studying alone. To transform not only learning functionalities but also the mindset underlying digitalization and data requires at least two semesters, aiming to eliminate preconceptions about digital technologies and lower the barrier of digital functionalities.

*4.3. Interview Analysis*

4.3.1. Preconceptions of Humanities Major Students

Students belonging to the so-called "digital generation" appeared to have either never contemplated or had very limited knowledge about the concept of convergence. Their exposure to computer education opportunities seemed to be relatively scarce. EdTech was unfamiliar and somewhat daunting to them. During their engagement with technology, curiosity, and confusion appeared to coexist. Consequently, their response to the integration of technology into humanities majors was marked by a mixture of curiosity and skepticism. This entrenched thought process among the digital generation created a significant barrier to digital transformation in education. It seemed like students perceived technology and humanities as two polar opposites that could never intersect. More than external factors, internal biases appeared to play a greater role in making EdTech seem challenging to these students. The hurdles they faced in embracing EdTech were largely constructed within themselves even before encountering the technology. This study demonstrated that, among digital generation university students, there had been limited opportunities for digital education during their middle and high school years, and these limited opportunities continued into their time in university.

4.3.2. Enjoyment

At some point, students discovered the driving force behind their newfound enthusiasm for the course. This moment marked a creative turning point. Students found pleasure and satisfaction in utilizing domain knowledge fed into the data to create something. At this juncture, they gained confidence in tandem with a shift in perception. Once they overcame their psychological thresholds via a few attempts, they savored the narrative essence of EdTech. The course content should not merely focus on the theoretical and obscure aspects, such as the workings of computers or the algorithms' intricacies. It should provide a challenging yet enjoyable experience, with the immediate application of acquired knowledge being paramount. Furthermore, the applicability of what they learned was of utmost importance. Students seemed to relish the thrill of instant implementation. As the semester progressed, students appeared to lose their initial hesitations and developed a sense of adventure toward the creative utilization of technology. For some, the 3D modeling course became a source of healing and happiness. Utilizing 3D modeling naturally facilitated gamification effects within EdTech.

4.3.3. Frustrations

Most humanities university students have grown weary of text-based learning. Alternative curricula should be developed to replace this educational method. Students consider EdTech as both a method and a goal in education. In higher education, EdTech is scarcely used, and in almost all courses, EdTech is not utilized. Even in cases where it is used, the educational component is lacking, and it is predominantly a tool for technology. Therefore, students tend to prioritize skills aimed at employment over those they can apply to their own studies.

### 4.3.4. Employment

While these students belong to the digital generation, their strong connection with the humanities is apparent. Consequently, they often approach EdTech with bewilderment and associate technology primarily with employment prospects. This is the dilemma for humanities majors. They recognize the necessity of being tech-savvy to thrive, but mastering technology is not their forte. Students mentioned that the course broadened their perspective on technology, implying an expansion of career options.

### 4.3.5. Perception Shift

Students who participated in the "3D Time Machine" course applied their newfound knowledge to other assignments. This development is noteworthy. Edtech, with its emphasis on software education, holds the potential for widespread adoption. Initially, students learn technology as a skill, but over time, it becomes part of their skill set. As a result, they independently seek applications for their acquired knowledge and hone their skills to achieve better outcomes. Hardware-centered EdTech enhances the efficiency of instrumental learning environments. In contrast, software-centered EdTech nurtures the cultivation of digital proficiency, encompassing a mindset and disposition toward technology, consistently fostering a sense of purpose in its utilization.

Although it might be assumed that these digital natives would have a natural affinity for technology, interviews with students revealed an initially vague and detached attitude toward technology. However, once they began participating in EdTech courses, they found them engaging and enjoyable. The issue lies not in EdTech itself but in how it is used solely for knowledge acquisition. A more effective form of EdTech involves the learners in creating tangible outcomes [34]. In such instances, students show interest in understanding how the technology used in EdTech is applied in the industry.

While the humanities students naturally stated that they had no prior opportunities for programming, they expressed satisfaction with narratives that combined knowledge and skills [35]. Despite the difficulties and inconveniences associated with learning new technology, they derived fulfillment from integrating knowledge and technology into a narrative [36]. The use of the term "satisfaction" beyond mere enjoyment implies a sense of pride in overcoming challenges and gaining confidence. Their expanded perspective on the world suggests not just a broadening of career choices but also an opportunity to discard preconceived notions and biases. The union of data and EdTech has the capacity to bring about transformative changes in individuals. It not only leverage the efficiency of physical technology for knowledge acquisition but also allows individuals to adopt the perspective of a creator using technology to produce content [37].

### 4.3.6. Sustainable Education with EdTech

The students took pride in the fact that they were creating something in the realm of digital "technology." As humanities majors, they initially accepted technology as something beyond their ability to realize, thus maintaining a passive attitude that separated them from the digital domain. Concepts like digital, computer, and technology seemed to have settled in their minds as forms of resignation and self-pity.

EdTech is about using technology for education, not teaching technology. Therefore, it is essential to quickly familiarize students with easy-to-use and versatile technologies. To live as responsible citizens in the future, students must be taught technologies they can handle independently and proactively. The students who participated in the "3D Time Machine" expressed considerable regret that the class only lasted one semester. They desired to learn technology that goes beyond a mere hobby, something that can practically support their studies, career choices, and lives. Via the class, they managed to alleviate some of the fear and vagueness; actually, some students became too serious and hesitated in operating some interface commands associated with the digital world, gaining the courage to confront the digital realm, which is the most significant achievement brought

by this class. This is because it provides a foundation for students to study continuously and independently.

They consistently expressed their willingness to study, even if they must do it alone. This implies the need for introducing a sustainable form of lifelong learning in education. To enhance practical abilities for sustainable development, schools should prevent citizens from becoming isolated and enable them to actively participate in society. A society primarily centered on virtual worlds based on digital data will soon emerge, and polarization will occur based on participation in various activities within these virtual worlds.

If humanities majors like the students in the class are not provided opportunities to achieve something in the field of technology, there will be an imbalance in society. Considering the rapidly changing digital industry, university students entering society and businesses must have easy access to and be able to learn software-centered EdTech. In essence, when sustainable education is provided to citizens, a sustainable world can be created.

## 5. Discussion and Conclusions

Via this study, it has been substantiated that opportunities for digital education are indeed limited for the digital generation. Moreover, even when such opportunities exist, humanities-major university students perceive them as exceedingly challenging. These educational approaches and contents require improvement. While the role of educators may vary, it is imperative that digital-related courses in the curriculum align with the foundational competencies of students. Although these students are at the university level, their digital literacy remains at an elementary school level. This situation can be attributed to the Korean educational system, which, due to the emphasis on college entrance exams beginning in middle school, does not require students to study subjects unrelated to these exams. In such a context, if university students are simply taught digital competencies with an assumption that they are as adept as adults, it not only fails to lower the entry barrier to digital transformation education but raises it higher [34].

There are two potential solutions to this issue: either introducing digital subjects into the college entrance exams or ensuring that students acquire digital competencies well before entering middle school. Currently, Korean university students are challenged to teach digital subjects. Once they enter university, they must decide on their future career paths, and society expects them to possess digital competencies. However, in order to gain admission to university, they have received no preparation in terms of competencies needed for the future society [15]. The EdTech curriculum at universities that caters to these digital generation university students should be composed of fundamental courses designed to be engaging and allow students to participate without reservation. Furthermore, the outcomes of these courses should be capable of influencing their future career choices. There is a need for an educational buffer zone to allow an engaging approach to digital technology. This buffer zone should be filled with software-centered EdTech.

Students who enrolled in the "3D Time Machine" course testified that they gained confidence in digital literacy via an easy and engaging curriculum. They felt that their digital competencies had significantly improved. This is the realization of a buffering effect in digital education. In the "3D Time Machine" course, students acquired data literacy and digital literacy using 3D modeling software, enabling them to unfold their own worldview in the digital realm.

One of the most striking findings from this study gathered via feedback from the digital generation university students, was their substantial skepticism regarding the current educational paradigm. Until now, all education has been centered around acts derived from "writing." Although today's university students are often referred to as a generation comfortable with digital technology, their actual application of digital technology in education is quite limited. Even if we stretch the definition to include note-taking on tablet computer screens and using the internet during classes, it is clear that educational activities predominantly reliant on the millennia-old "writing" paradigm have not changed. Consequently,

despite living in the digital age, these students are still living in an analog world when it comes to activities aimed at achieving educational goals. When transitioning to digital education, it is essential for it to have characteristics distinct from analog education, and the impact it has on learners' cognition must differ from the impact seen in analog education. The digitization issue is not solely about efficiency enhancement; it demonstrates how critical it is to foster critical thinking and new levels of cognitive abilities in the entities that utilize digital tools.

The "3D Time Machine" course intended to shift the educational paradigm by emphasizing a "hands-on" approach, which turned out to be highly effective. Students all found satisfaction and interest in witnessing their creations come to life at their fingertips. Moreover, there was discussion regarding the teaching style that educators employing software-centered EdTech need to adhere to. Surprisingly, a significant number of today's university students, despite being referred to as the digital generation, have never even tried Photoshop. When teaching students at this level, educators must lower their expectations and establish an atmosphere of friendly guidance, showing easily comprehensible outcomes in advance. During the lectures, some students might require detailed instructions on even the most basic keyboard shortcuts. The reason for using EdTech is not to transform learners into professional technicians, so fast-paced lessons may not be ideal. Instead of keeping pace with a fixed curriculum, it is more appropriate to adapt the course's progression according to the learners' comprehension level. Furthermore, educators should possess a significantly higher level of technology literacy and data literacy compared to the learners. This allows the educational goals of EdTech to extend beyond merely teaching software usage to potentially educating within metaverse environments.

Education for sustainable development fosters environmental friendliness, health, inclusiveness, equality in the world, nation, and city, work, employment, entrepreneurship, and civic spirit, and provides lifelong learning opportunities for all age groups and all educational levels, including those residing in the country and local communities, not just university constituents. To achieve this, SDG education at universities must encompass a wide range of approaches, including not only majors and liberal arts but also science and technology, family, employment, industry and economic development, immigration and integration, civic awareness, social welfare, and public finance-related policies [2]. Above all, student education that enables them to survive in the digital environment is essential, and humanities majors may require even more careful digital education, as they might be vulnerable in this field.

At present, educational institutions carry a substantial societal obligation to produce learners capable of actively engaging in the data era and the digital landscape. Digital technology has become an essential survival skill, much like the traditional imperative education in reading and deciphering text, enabling the recreation of experiences in the virtual realm, the new frontier of human existence. Curricula incorporating digital technology should transcend mere mechanical training, focusing on fostering creativity and multidimensionality. Digital data narratives are an educational approach that combines environmental factors, social factors, and governance elements to not only enhance competencies centered around digital data but also internalize competencies related to the SDGs [38].

The "3D Time Machine" course, developed based on a digital data narrative instructional model, is a metaverse course composed of three elements: cognitive psychology, design thinking, and digital data narrative. It not only aims to reduce learners' vague fears about the digital data society, preparing them for active participation in the metaverse, using the 3D modeling tool Blender but also to serve as a software-centered EdTech curriculum that enables the expression of creative imagination. The purpose of the "3D Time Machine" course is twofold. Firstly, it encourages learners to increase their digital data literacy, enabling them to witness the results of their creative thoughts. Secondly, it lowers the barriers of entry for non-digital data major students, allowing them to venture into the

digital data domain. Many students who took the "3D Time Machine" course switched their career paths.

To sum up students' evaluations, the most prominent observation is that they found the "3D Time Machine" course extremely interesting. They experienced enjoyment and satisfaction in the realization of their thoughts and were engaged in the process. Although they initially faced challenges in 3D creation, as they became familiar with the functions of 3D modeling engines, they found satisfaction in their ability to produce new things. Moreover, students appreciated the freshness of this unique educational pattern compared to traditional history courses. Importantly, it broke down their self-imposed barriers, allowing humanities majors to explore various new possibilities in academics and career paths. The "3D Time Machine" course brought a transformation to historical data utilization in the metaverse, offering students a novel perspective.

Via this study, it became evident that practical digital education beneficial to students should be purposeful and applicable to their future careers. It should offer enjoyable outcomes while fostering a sense of accomplishment, thereby stimulating the students' cognitive abilities to perceive the world. When such considerations are considered, and the "curriculum development of an EdTech class utilizing 3D modeling software" succeeds, the university evolves into an institution capable of fulfilling a sustainable obligation. It becomes adept at fostering a genuine digital generation equipped with sustainability, fulfilling its ongoing duties effectively.

In the future, we anticipate the emergence of interoperable virtual world education environments. Students who possess 3D modeling skills will have a digital writing tool at their disposal, enabling them to express their worldviews in any metaverse. From an educator's perspective, 3D engine-based metaverse content courses mark the initiation of EdTech evolving into "TechEdu" (Technological Education). While technology has thus far augmented education to enhance efficiency, these 3D content creation courses, designed with metaverses and sustainability in mind, indicate technology's role in education is changing [36]. For instance, technology enables the real-time and seamless implementation of participants' thoughts, allowing for far more creative communication than traditional explanations. The "3D Time Machine" course was planned and launched with this vision. However, evaluating students' cognitive development requires in-depth research and separating the "3D Time Machine" course into basic and advanced levels with curated curricula. Two semesters are insufficient for a thorough analysis of student capabilities.

**Author Contributions:** Writing—original draft, W.C.; Writing—review & editing, S.K. All authors have read and agreed to the published version of the manuscript.

**Funding:** This research received no external funding.

**Institutional Review Board Statement:** Not applicable.

**Informed Consent Statement:** Not applicable.

**Data Availability Statement:** No new data were created or analyzed in this study. Data sharing is not applicable to this article.

**Conflicts of Interest:** The authors declare no conflict of interest.

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
