# Peer review of "Curriculum Development of EdTech Class Using 3D Modeling Software for University Students in the Republic of Korea"

_sustainability, doi:10.3390/su152416605_

Round 1

Reviewer 1 Report

Comments and Suggestions for Authors

The chosen theme has aroused my interest and is also topical.

The literature study is comprehensive, relevant, and well-connected in content. Very well-aized link between the goal and objectives of sustainable development.

The method is well described, it is clearly distinguished where, how, and when everything takes place. 

As for the results, I can say that the description of the rooms for the theme "Chunhyang's Room" is impressive.

The interviews of the students highlight the importance and attractiveness of the course conducted, especially as some of them were unsure about the choice.

The analysis of the interviews is also relevant, highlighting the important aspects of what the students said based on their experience of taking the 15 courses.

The implications are well drawn, as are the conclusions.

Author Response

Thanks for your review and helpful comments.

Please see the attachment, we attached 'the revised manuscript' and 'revision report.'

Reviewer 2 Report

Comments and Suggestions for Authors

The research is very valuable, especially in that it highlights that often the students' digital proficiency is not that good at all, and it also deals with how to encourage students, who are non-specialists in the digital and data fields, to learn use technologies, because they are part of our century and future.

There is a reference to the German Rectors' Conference, but is it a global call? This should somehow be more intertwined with other findings in the theoretical part.

The content of the study course, within the framework of which the research was conducted, is very culture-specific (I mean exactly the content of the students' choice of topics), so the article would require more of its learning context and its importance in the overall learning outcomes, in the acquisition of qualifications.      

It is interesting to read the student interviews wlogs, but it would not be bad if those data were processed with some qualitative research tool, then the data presentation was supplemented with student statements.

The methodology part should explain the methods of data analysis more.

Author Response

(The authors gave the same response as above.)

Reviewer 3 Report

Comments and Suggestions for Authors

1. Objective analysis

The title of the article is Development of a Digital Data Narrative EdTech Model: 3D Modeling Software Class for University Students in Republic of Korea, however the objective of the study is “This research aims to elucidate to what extent the generation commonly referred to as " digital natives" - the university students of today in 2023 - are utilizing digital technology in their learning processes and what factors should be considered when designing effective curricula for them.”

It guides us to know two aspects:

to. If university students use digital technology in their learning processes; and

b. Identify the factors that should be considered when designing effective resumes for them.

So where does the manuscript take us? To present the EdTech narrative model of digital data? Or to know if students use technology, how they use it, what attitudes they have towards it, etc., which would allow us to identify the factors or dynamic variables linked to the use of technologies in learning processes.

If the answer is to present the development of the EdTech model, the work must be reoriented towards the presentation of this model, what it proposes, what parts it has, what theories it is supported by, how it has been validated, etc. Etc.

If the answer is to know if students use technology and identify the factors or dynamic variables linked to the learning processes, the title of the manuscript must be evaluated and its construction must be adapted to it.

It is suggested: Make an effort to standardize an objective that can be achieved after analyzing the results, concluding on aspects related to it. To the extent possible, use the same objective when referring to it within any part of the manuscript.

2. The topic of presenting a proposal for the development of the EdTech model or knowing if students use technology and identifying the factors linked in the learning processes, constitutes a current and original topic that can help researchers and professionals around the world.

It is suggested: support the work using references from publications in indexed journals, correctly identified (DOI or URL) that help provide the corresponding scientific basis.

3. The contribution of the manuscript, depending on the definition of its objective, would be aimed at proposing an EdTech model that could be used to generate learning or identify aspects that those responsible for universities could use to improve their curricula.

It is suggested: Make an effort to review the complete coherence of the manuscript.

4. Regarding the Method:

- Better define the sample and the inclusion or exclusion criteria were raised.

- Define the instruments that were used, how the information was collected and how they related the meanings and experiences of the participants.

- What type of analysis was used (statistics for qualitative research).

5. The results must show tables and figures that associate what was evaluated (for example, the identified factors) so that in this way the reality studied can be explained.

6. The conclusions must respond to what the objective seeks, in a concrete way, preferably without relying on references, since it is the contribution to the authors' knowledge.

7. Significantly increase the number of references obtained from indexed databases (Scopus and WoS) in the last 5 years, to provide scientific support to the manuscript.

8. Carry out an exhaustive review of the presentation of each reference, integrating the necessary elements (DOI preferably) to better evaluate the proposal.

Author Response

(The authors gave the same response as above.)

Reviewer 4 Report

Comments and Suggestions for Authors

The research focuses on experimenting with an Educational Technology (EdTech) classroom model based on software, implemented through the course "3D Time Machine" at a South Korean university. Specifically, the course taught 3D modeling with Blender for history learning, equipping students with digital skills and promoting storytelling from "their unique perspectives". Interviews with the students revealed that they overcame the "fear" of digital technology (we might question whether, instead of "fear," in the context of this digital generation, we should talk about insecurities common to any student in any sociotechnological context).

The theoretical framework is comprehensive and up-to-date, combining the contributions of various authors with numerous interesting facts related to the integration of technology in educational settings. Furthermore, the authors do not avoid addressing the contradictions that technology can potentially generate in educational practices. However, the selected literature hardly includes any studies on the application of new technologies to the teaching and learning of history. Additionally, perhaps the self-citations by Choi appear excessive.

In section 1.3, the problem driving the research is clearly specified: the competencies of the digital generation of students are not significantly different from those of analog generations. This justifies the adoption of a rather ambitious objective: "to provide insight into what digital education means to this generation, what aspects they value, what they wish to learn, and how they view digital education and digital learning environments," as well as "providing an opportunity to assess what EdTech should encompass in the context of next-generation digital literacy education in virtual environments."

As for section 1.4, it appears to be insufficient, as instead of explaining the strategies for data collection and analysis, it presents a kind of summary of the report and a characterization of the participants. It is proposed to expand this section further and include the questions asked during the interviews with the students.

The proposal of the "3D Time Machine" course, around which this study revolves, is well-described. It constitutes an educational experience based on a fairly comprehensive specific curriculum, including the use of 3D design software, along with the necessity to export this design to a metaverse platform. Both the tables and images in the article are appreciated.

Regarding the organization of the article, I believe that section 1.4 should have greater prominence, constituting section 2. Within this section, the description of the "3D Time Machine" could be included as part of the research methodology. From 2.3 onwards, the text's organization becomes particularly confusing. Section 2.3 addresses "results," distinguishing this information from the interviews, as if the participants' testimonials were not considered results of the research... In this regard, I recommend consolidating all the information obtained from the experience into a single section titled "Results Analysis," without the need to reproduce the complete "interview logs", but integrating them in the analysis. Similarly, a new section titled "Discussion and Conclusions" could be created, combining sections 3.2.7 and 4, perhaps incorporating more dialogue with the conclusions drawn by other authors in their respective studies.

In conclusion, I believe that the article provides a detailed description of the development of a highly interesting and innovative educational experience, which undoubtedly offers a new and well-justified perspective on the importance of promoting sustainable technological integration, in this case, among university students. However, I think that the previous recommendations can help to more clearly convey the article's significance. Mainly, I suggest reviewing the references, expanding the sections on the methodology used and the discussion conducted, respectively, and reorganizing the text for greater clarity and impact.

Comments on the Quality of English Language

 I recommend reviewing the wording to refine some strange sentences (e.g., “Empirical evidence confirming that distinguishing generations based on digital activity is a valid method for discerning whether digital activities have shaped a new generation”).

Author Response

(The authors gave the same response as above.)

Round 2

Reviewer 3 Report

Comments and Suggestions for Authors

Thank you very much for raising your observations. We will be attentive to your future investigations.